# Inflammatory Pseudotumor of the Liver or Intrahepatic Cholangiocarcinoma, That’s the Question: A Review of the Literature

**DOI:** 10.3390/cancers16172926

**Published:** 2024-08-23

**Authors:** Matteo Barabino, Gaetano Piccolo, Andrea Tramacere, Stefano Volponi, Claudia Cigala, Umberto Gianelli, Carla Codecà, Francesca Patella, Giorgio Ghilardi, Francesca Lecchi, Paolo Pietro Bianchi

**Affiliations:** 1FACS, General Surgery Unit, Department of Health Sciences (DISS), University of Milan San Paolo Hospital, Via Antonio di Rudinì 8, 20142 Milan, Italy; gaetano.piccolo@asst-santipaolocarlo.it (G.P.); andreatramacere9@gmail.com (A.T.); stefanovolponi99@gmail.com (S.V.); giorgio.ghilardi@unimi.it (G.G.); fra.lecchi@gmail.com (F.L.); paolopietro.bianchi@unimi.it (P.P.B.); 2Pathology Unit, Department of Health Sciences, University of Milan San Paolo Hospital, 20142 Milan, Italy; claudia.cigala@asst-santipaolocarlo.it (C.C.); umberto.gianelli@asst-santipaolocarlo.it (U.G.); 3Division of Oncology, San Paolo Hospital, 20142 Milan, Italy; carla.codeca@asst-santipaolocarlo.it; 4Department of Diagnostic Radiology, San Paolo Hospital, 20142 Milan, Italy; francesca.patella@asst-santipaolocarlo.it

**Keywords:** inflammatory pseudotumor of the liver, intrahepatic cholangiocarcinoma, liver surgery, differential diagnosis, liver imaging

## Abstract

**Simple Summary:**

Rare diseases represent a significant health problem since patients face difficulty obtaining a rapid diagnosis and a proper treatment. An inflammatory pseudotumor of the liver (IPTL) is a rare and benign entity in which reaching a correct preoperative diagnosis can be challenging since it is similar to the worst form of liver cancer, intrahepatic cholangiocarcinoma (ICC). Our paper aims to report our experience and to review the available literature on this topic, thus summarizing previous experiences and central issues to point out a prompt road map of treatment that still needs to be standardized. IPTL is not associated with substantial risk factors and presents with faint symptoms. Imaging data via MRI and CT scan are not specific, thus often requiring ultrasound (US)-guided biopsy. Proper and widely accepted gold standard treatment does not exist; conservative strategies represent an accepted option, while the decision for surgery still exists where there is a suspicion of malignancy.

**Abstract:**

An inflammatory pseudotumor of the liver is a rare tumor-like lesion composed of polymorphous inflammatory cell infiltrates and variable amounts of fibrosis that can often mimic a malignant liver neoplasm. The etiology of inflammatory pseudotumors of the liver is unknown; symptoms are faint and imaging non-specific. Thus, it is often hard to make a diagnosis preoperatively and it is not so rare to over-treat patients with this disease or vice versa. Thus, more profound knowledge is necessary to plan appropriate disease management. We reported our two cases and systematically searched the literature regarding IPTL. We selected articles published in English from four databases, PubMed, Scopus, Web of Science and Google Scholar, and we included only articles with consistent data. Twenty nine papers fulfilling criteria for the review were selected. The analysis of 69 cases published from 1953 confirmed that the risk factors are unclear, the imaging data is not specific, and biopsy is crucial but not so widely used in clinical practice due to the procedure’s related risks, and relatively low effectiveness and improvement in imaging analysis. Regarding treatment, surgeons have moved towards a more conservative attitude over the years due to better imaging quality and patient surveillance. However, surgery remains the modality of choice for most cases with an indeterminate diagnosis. Even if an inflammatory pseudotumor of the liver is a benign tumor with a good prognosis, not requiring any treatment in most cases, sometimes it remains challenging to differentiate it from ICC; therefore, there is a solid recommendation to manage this condition with a multidisciplinary team.

## 1. Introduction

Hepatic inflammatory pseudotumor (IPTL) is a benign, non-metastasizing tumor characterized by the presence of myofibroblastic spindle cells associated with heterogeneous populations of inflammatory cells, such as plasma cells, lymphocytes, and macrophages, as well as areas of fibrosis and necrosis without cellular anaplasia or atypical mitoses [1]. A hepatic inflammatory pseudotumor of the liver (IPTL) was described first by Pack and Baker in 1953 [1], including a heterogeneous group of mass-forming benign lesions involving many organs, of which liver is the organ in 8% of cases [2]. Its deceptive name “pseudotumor” was given by Umiker in 1954 as the ability in mimicking a malignant tumor on imaging, especially peripheral intrahepatic cholangiocarcinoma (ICC) [3,4,5]. IPT has been reported in 0.7% of all liver mass, thus it is a very rare benign lesion often disappearing spontaneously [6]. Hepatic IPTL can be classified into two types: fibrohistiocytic and lymphoplasmacytic. Fibrohistiocytic IPTL is characterized by xanthogranuloma, multinucleated giant cells, and neutrophilic inflammation, while lymphoplasmacytic IPTL shows diffuse lymphoplasmacytic (IgG4-positive plasma cells) and characteristic eosinophilic infiltration [7]. These two groups differ also in terms of clinical features, such as average age, tumor size, and location.

The lymphoplasmacytic type is associated with IgG4-related disease (IgG4-RD) according to the 2020 revised comprehensive diagnostic criteria for IgG4-RD [8].

The fibrohistiocytic type is observed most commonly in the peripheral hepatic parenchyma, resembling mass-forming (MF) intrahepatic cholangiocarcinoma (ICC). On the other hand, the lymphoplasmacytic type is shown near the hepatic hilum, mimicking the periductal infiltrating (PI)-type of intrahepatic cholangiocarcinoma (ICC) [7]. For these reasons, IPTL often requires a liver biopsy, even if the histological criteria alone may sometimes be insufficient to diagnose with certainty. Consequently, surgery has been suggested in many cases as the treatment of choice for patients with indeterminate diagnoses [9]. This review aims to collect and analyze all the available literature systematically, then summarize and discuss the evidence on IPTL.

## 2. Materials and Methods

### 2.1. Patients

Between 2022 and 2023, we identified, in our surgical department (San Paolo Hospital, Milan, Italy), 2 patients with a hepatic inflammatory pseudotumor of the liver (IPTL) that underwent ultrasonography, a computed tomography (CT) scan, magnetic resonance imaging (MRI), and an ultrasound (US)-guided needle biopsy to obtain a diagnosis. The decision-making process was based on a multidisciplinary team (MDT), including a liver surgeon, a hepatologist, a radiologist, and an oncologist. In the case of surgery, the pathologist confirmed the histology, while after conservative treatment, the patient underwent a long-term follow-up.

### 2.2. Literature Review

A systematic search of the literature, up to March 2024, was performed in four electronic databases (PubMed, Scopus, Web of Science, and Google Scholar) to identify papers of interest for this systematic review. The search included the following keywords and medical subject heading terms, alone or in combination: “hepatic inflammatory pseudotumor” and “inflammatory pseudotumor of the liver”. The article search was carried out according to the PRISMA (Preferred Reporting Items for Systematic Reviews and Meta-Analyses) statement [10], as presented in Figure 1. As inclusion criteria, the papers had to be written in English with the full text ascertainable and to contain the necessary data we were searching for (symptoms, size and localization of the lesion, treatment, outcome, follow-up). Finally, they had to be about IPT only. Additionally, we scrutinized the references of reviewed articles to obtain any other reference that eluded the primary search. We examined all identified articles and reviewed their reference lists to include other potentially relevant studies. Two independent authors reviewed the studies (AT and SV) for inclusion. We decided on the final inclusion after a detailed examination of all the manuscripts.

## 3. Results

### 3.1. Cases Presentation

Both of the two patients were female (59- and 57-years-old) without any significant past medical history and were referred to our hospital for abdominal pain. The first case presented symptomatic gallstones disease without any sign of cholecystitis or cholestasis, while the second showed a mild epigastric abdominal pain associated with recurrent diarrhea. Liver blood tests revealed a mild increase in transaminases with normal white blood cells, PCR, and serum tumor markers in the first case, while in the second one all exams were within normal limits.

### 3.2. Imaging

#### 3.2.1. Case 1

The abdominal ultrasound mentioned multiple small gallstones with initial wall thickening and without biliary tree dilation. Next to the gallbladder in S5, a 35 mm hypoechoic mass was revealed. Gd-EOB-DTPA-enhanced magnetic resonance imaging (MRI) showed a 40 mm tumor with a low intensity on the T1-weighted images, a light high intensity on the T2-weighted images, and a low intensity on the hepatobiliary phase, highly suspected to be ICC (Figure 2).

The chest and abdominal enhanced CT scan with 3D reconstructions (Figure 3) confirmed a nodule with an increasing enhancement in the portal phase, thus confirming a suspect of malignancy but without any sign of vascular infiltration, distant metastases, and/or associated lymphadenopathy or cirrhosis.

Nevertheless, imaging data favored ICC, so we performed a percutaneous ultrasound (US)-guided tumor biopsy to make a definitive diagnosis. The pathology with hematoxylin and eosin staining revealed no evidence of tumor cells. Severe fibrosis and inflammatory cells infiltration were observed, in particular plasma cells (CD 138+, polyclonal kappa and lambda chains).

#### 3.2.2. Case 2

The abdominal ultrasound showed a 36 × 28 mm hypoechoic lesion in S5 near to a normal gallbladder. The Gd-EOB-DTPA-enhanced magnetic resonance imaging (MRI) showed a 38 mm tumor in segment 5 with a low intensity on the T1-weighted images, and a light high intensity on the T2-weighted images and DWI (Figure 4). The abdominal enhanced CT scan (Figure 5) confirmed a nodule with enhancement in the arterial phase and washing out in the portal, which is highly suspect for mass forming cholangiocarcinoma versus fibro-lamellar HCC within a normal liver.

An ultrasound (US)-guided tumor biopsy revealed, at pathology with hematoxylin and eosin staining, inflammatory cells infiltration with giant cells and plasma cells (CD 138+, polyclonal kappa and lambda chains, no IgG4 expression). Finally, no tumor cells or any cytological atypia were revealed (Figure 6).

### 3.3. Treatment

#### 3.3.1. Case 1

According to the biopsy, the patient was scheduled for robot-assisted cholecystectomy (CMR Versius Surgical Robotic System, Versius^®^). Still, intraoperatively, a highly suspicious feature of an ICC tumor infiltrating the gallbladder was revealed without any clear dissection plane with the duodenum (Figure 7); thus, after proper informed consent, a non-anatomic open S4b-S5 resection en bloc with the gallbladder and marginal duodenal resection, plus lymphadenectomy, was performed.

The post-operative course was uneventful and histologically the tumor was characterized by a heavy inflammatory infiltrate consisting of plasma cells and lymphocytes with no evidence of malignancy. The final pathology report revealed an inflammatory pseudotumor (IPT) of the liver (Figure 8).

#### 3.3.2. Case 2

The patient was scheduled for conservative treatment with prednisone 75 mg/die for 1 month, followed by a gradual reduction in the therapy dose, thus obtaining almost the complete disappearance of the liver mass pseudotumor, measuring from 38 mm to 8 mm after 20 months of follow-up.

### 3.4. Literature Review

#### 3.4.1. Patient’s Characteristics

At the time of this review, a total of 71 cases have been described in 29 papers, including our two cases [1,11,12,13,14,15,16,17,18,19,20,21,22,23,24,25,26,27,28,29,30,31,32,33,34,35,36,37,38,39]. This represents the largest published literature review to date on this rare liver disease. A summary of the clinicopathological features of all cases is shown in Table 1.

The mean age was 43 years (range, 2 to 83 y), including 20 pediatric and 49 adult cases. Most of the pediatric patients were female (54%), while males were predominant in adulthood (65.5%). According to our review, no specific risk factors have been identified in a pediatric setting, even if some author reported a correlation with severe congenital neutropenia (Kostmann’s disease) and recurrent episodes of bacterial infections [23].

In adult patients, biliary disease does not seem to represent a significant risk factor for the development of the hepatic inflammatory pseudotumor (17%), contrary to other authors reporting a percentage up to 65.2% [6]. In twenty percent of the patients, the inflammatory pseudotumor of the liver was associated with IgG4-related disorders involving various organs, including the pancreas (autoimmune pancreatitis), salivary glands, and liver [10,39,40]. Isolated IgG4-RD confined to the liver is rare, and most of the articles included in the present review do not mention this item (56/71, 78.9%), especially before 2012 [41]. Based on this review, the majority of the patients were symptomatic (83.1%); 37 and 38 of these presented, respectively, abdominal pain (53.5%) and fever (52.1%). Only nine patients developed jaundice (Table 2).

#### 3.4.2. Diagnosis

Laboratory exams are not useful for guidance when IPTL is suspected, except for cholestasis or increased transaminase in coexisting complicated cholelithiasis (12.7% of cases). Moreover, serum tumor markers are useless, because alpha fetoprotein (AFP) was almost within normal limits in the reported cases. CA 19.9 was not measured in 38 out of 71 patients (53.5%), was shown to be increased in 8 (11.3%), and was within normal limits in 2 cases. Finally, in the present review, the IgG4 plasma levels were unavailable in 63 out of 71 cases (88.7%), thus only routinely appearing as a measurement after 2009. Non-invasive diagnosis is often difficult, since the liver’s IPT on imaging may show different behaviors, varying from progressive contrast enhancing (CE) in the portal phase during the CT scan or at hypo-intensity in the T1 phase, hyper-intensity in the T2 phase, and hypo-intensity in the hepatobiliary phase at MRI, mimicking an intrahepatic cholangiocarcinoma (ICC), to a typical CE wash-in and wash-out in the vascular phases at CT or MRI, which is suggestive for hepatocellular carcinoma [6]. Nevertheless, a CT scan may reveal different patterns of contrast enhancement depending on the grade of IPT’s fibrosis, being generally widespread and located in the center of the tumor, thus relegating vascularization in the periphery, mimicking ICC or necrotic hepatocellular carcinoma or colorectal liver metastases. In the case of indeterminate diagnosis, ultrasound (US)-guided biopsy was carried out in 38% of cases. In the present review, the mass showed to be solitary in 84.5% of cases with a variable size ranging between 7 and 200 mm (mean 58.7 mm) with a predilection for the right lobe (64.8%), as in our two cases.

#### 3.4.3. Treatment and Clinical Course

Since the first case was described ninety years ago, it is understandable there has been a deep change in approach towards different eras (Table 1). Thus, we decided to split patients into two groups, before and after 2000, as reported in Table 3.

Surgery was effective in 43 out of 45 patients (95.6%), with a 4.4% mortality (two case) due to post-operative complications. For the conservative approach, a complete or partial mass regression was revealed, respectively, in 16 (55.2%) and 5 cases (17.2%), while in 3 cases a failure required surgery (10.3%). Antibiotics and corticosteroid were the most used therapy. The maximum follow-up found was 192 months. During follow-up, six patients died due comorbidities or overcoming other tumors.

## 4. Discussion

A hepatic inflammatory pseudotumor (IPT) is a rare benign tumor-like lesion; there are many theories regarding the etiology of this disease, including bacterial and viral infections or autoimmune diseases. IPTs are often misdiagnosed as malignant tumors, such as intrahepatic cholangiocarcinoma (ICC) or hepatocellular carcinoma (HCC). According to our review, preoperative diagnosis is challenging, with liver tumor markers (AFP and CA 19.9) almost within normal limits in the reported cases. There were only rare cases of IPT with elevated serum levels of AFP [39,40]. In these cases, hepatic IPT mimicked an HCC and patients underwent hepatic resection. There are also no radiographic features for the diagnosis of IPT. The majority of patients underwent a CT/MRI scan. Fluorine-18 fluorodeoxyglucose positron emission tomography (FDG-PET) is not commonly performed; however, only a few IPT cases showed abnormal metabolic activity on FDG-PET [40,41].

Our cases revealed two opposite strategies in the treatment of IPTL in patients with mild symptoms, and quite similar features in CT/MRI and benign findings at biopsy, the former with liver resection and the latter with observation and medical therapy. Our dynamic strategy could be judged as not entirely acceptable, but it reflects general practice according to the present review. It is hard to draw a sharp and shared road map in front of such a rare event, which only accounts for 0.7% of all the liver tumors reported in 29 papers, including 71 cases with consistent data. This tumor-like lesion generally requires no surgical strategy since it has benign behavior. Nevertheless, pre-operative diagnosis is often challenging, even after MRI/CT scan imaging and/or liver biopsy.

A non-invasive diagnosis is often difficult since IPTL is rarely associated with cholelithiasis, a coexisting biliary tumor or an IgG4-related disease. Moreover, the symptoms are vague and imaging data need to be more specific to make a differential diagnosis without an invasive approach. In this regard, what we learned from this review is that some current authors preferred, in indeterminate cases, surgery first rather than ultrasound (US)-guided biopsy, as this was performed in more recent reports in 38% of the cases. The main explanation for this attitude is that biopsy remains a potential tricky maneuver, associated somewhat with a risk of bleeding, which may not always lead to a definitive diagnosis.

The treatment of IPTL remains controversial, oscillating between conservative and surgical attitudes depending on several factors that should be discussed within a multidisciplinary tumor board (MDT), including liver surgeons, medical oncologists, hepatologists, and radiologists. In our first case, we were scheduled to perform only the cholecystectomy, but the intraoperative finding changed our plan, directing us towards a liver resection associated with lymphadenectomy and partial duodenectomy. Finally, according to the pathological exam, we over-treated the patient, but it would have been quite impossible to act more conservatively in such an aggressive presentation. The critical point is to fully inform the patient preoperatively about the potential of a non-malignant outcome after surgery. Many authors reported a good result after conservative management with antibiotics, corticosteroids or non-steroidal un-inflammatory drugs, but some of these lesions recurred [9,42,43]. However, even though an IPT of the liver may spontaneously regress or decrease following antibiotic treatment, the common practice of excising a resectable liver tumor in the absence of a firm diagnosis sounds reasonable [44]. We noticed a real change in attitude before and after 2000 as described in Table 3, showing, through the years, a decrease in surgical approaches (from 88.5% to 48.9%) in favor of conservative approaches (from 11.5% to 51.1%), due to better imaging quality, clinical surveillance, and oncological knowledge. On the one hand, in front of any doubts of malignancy, liver resection is preferable because it brings the pathology diagnosis, thus avoiding the risk of a biopsy-related complication (dissemination or bleeding) and eliminating the possibility of an IPT recurrence. On the other hand, when an MDT hears of benignancy, we are authorized to keep on with the conservative strategy, followed by long-term US follow-up.

## 5. Conclusions

IPT remains a sneaky but rare tumor of the liver that must be differentiated from the malignant ICC.

Currently, even if we do not have so many strings to our bow to predict benignancy or to not suspect IPTL, it seems we are moving towards a more conservative attitude due to MDTs taking charge of the patient, the ability to carefully analyze patient’s medical history, sophisticated MRI and CT scan data, and at the same time, being able to ensure the long-term patient’s surveillance in the case of confirmed benignancy. We have to enhance the routine measurement of useful laboratory exams from CA 19.9 and AFP to IgG4 levels, even if they rarely make much difference, because any little clue may finally bring the truth. Moreover, we have to perform more biopsies in the face of consistent doubts, and even if they persist afterwards, surgery remains the modality of choice for most of the cases with an indeterminate diagnosis.

## Figures and Tables

**Figure 1 cancers-16-02926-f001:**
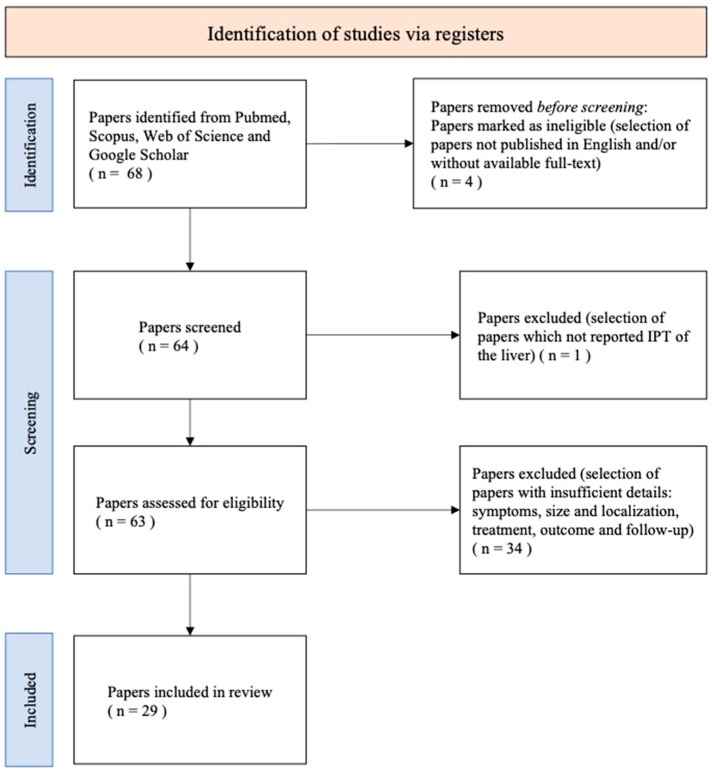
Flow chart summarizing the systematic literature review process.

**Figure 2 cancers-16-02926-f002:**
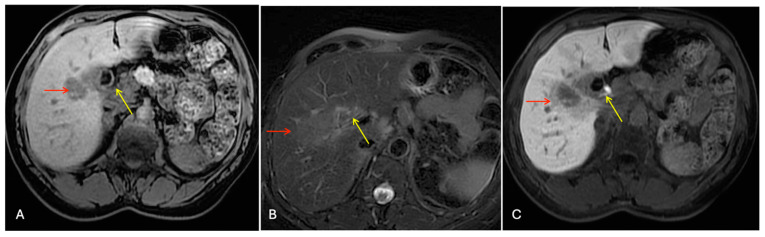
Sequential dynamic contrast enhancement liver MR images. Liver mass in segment 5 (red arrow), gallbladder (yellow arrow): (**A**) hypo-intensity in T1 phase, (**B**) faintly hyper-intensity in T2 phase, and (**C**) hypo-intensity in hepatobiliary phase.

**Figure 3 cancers-16-02926-f003:**
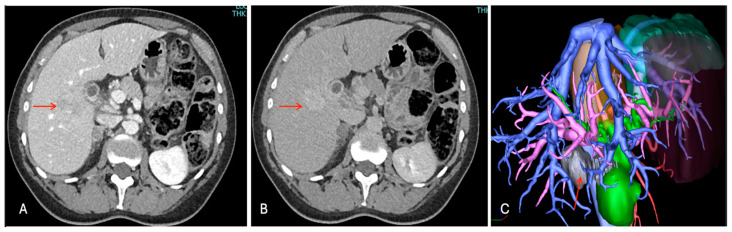
Contrast enhancement CT images with 3D reconstruction. Liver mass in segment 5 (red arrow), (**A**) iso-slightly hyperdensity in arterial phase, (**B**) progressive enhancing in portal phase, and (**C**) 3D reconstruction with relationship between tumor and vessels.

**Figure 4 cancers-16-02926-f004:**
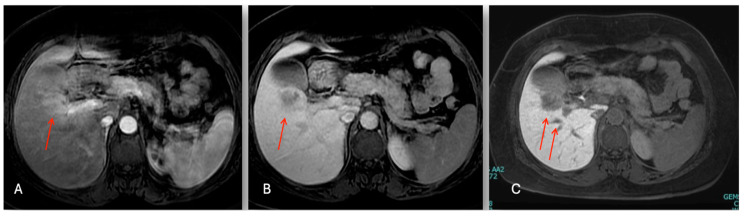
Sequential dynamic contrast enhancement liver MR images. Nodule in S5 (red arrow): (**A**) hyper-intensity in T2 phase; (**B**) hypo-intensity in T1 phase; and (**C**) hypo-intensity in hepatobiliary phase.

**Figure 5 cancers-16-02926-f005:**
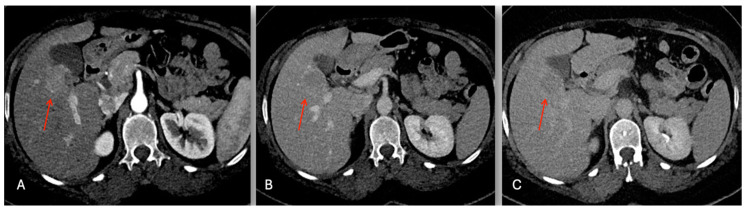
Contrast enhancement CT images. Liver mass in S5 (red arrow): (**A**) faint hyperdensity in arterial phase; progressive contrast enhancement wash-out in the portal (**B**); and delayed (**C**) phase.

**Figure 6 cancers-16-02926-f006:**
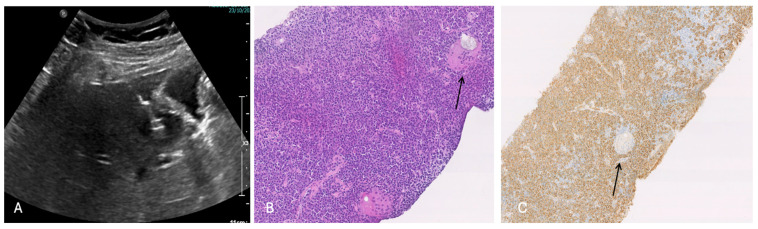
(**A**) Ultrasound (US)-guided biopsy of S5 nodule next to gallbladder; (**B**) pathological examination of the tumor showed fibrosis accompanied by lymphoplasmacytic infiltration (hematoxylin and eosin staining, low power × 100 HE); (**C**) Immunohistochemical staining shows inflammatory cells infiltration with giant cells (black arrow) and plasma cells (CD 138+, polyclonal kappa and lambda chains, no IgG4 expression) (low power × 100 HE).

**Figure 7 cancers-16-02926-f007:**
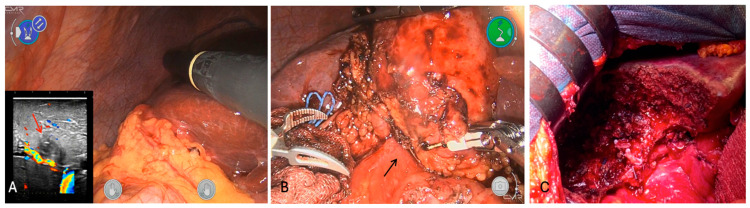
Intraoperative findings. (**A**) Hypoechoic nodule with a faint hyper rim in segment 5 next to P5-8 (red arrow); (**B**) no dissection plane between gallbladder and duodenum (black arrow); (**C**) final cut surface after S4b-S5 and partial excision of duodenum.

**Figure 8 cancers-16-02926-f008:**
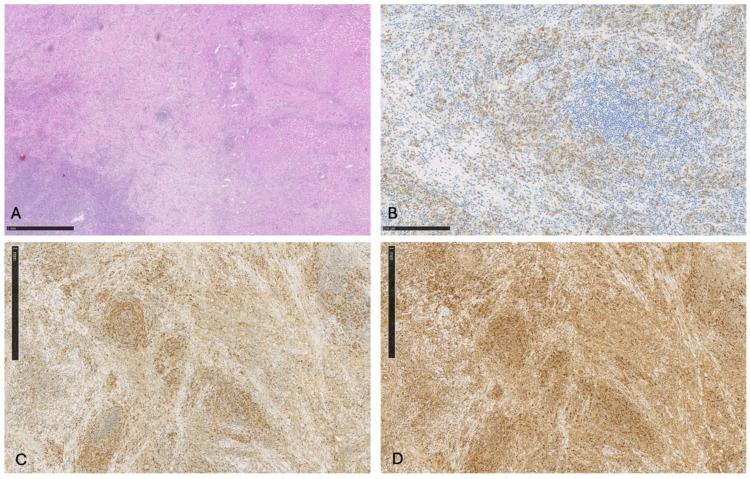
(**A**) Histopathological examination of hepatic IPT (hematoxylin-eosin staining, low power × 100 HE). Presence of a dense inflammatory infiltration: CD138+ plasma cells (high power × 400 HE) (**B**), polyclonal for Kappa (high power × 200 HE) (**C**), and Lambda (high power × 200) (**D**) mixed with granulocytes neutrophils and histiocytes.

**Table 1 cancers-16-02926-t001:** Summary of the clinicopathological features of all cases.

No.	Authors	Year	Patients	Sex	Age (Years)	Clinical Features	Underline Disease	Size (mm)	Location	Solitary or Multiple	Biopsy	Treatment	Outcome	Follow-Up (Months)
1	Present cases	2024	2	F (2)	57–59 (58)	AP (2)	CH (1)	35–36 (35.5)	R (2)	Solitary (2)	Yes (2)	Corticosteroids (1)	Disappeared (1)	18–20 (19)
2	Oh et al. [11]	2021	7	F (2)	46–83 (60.5)	AP (2), NS (5)		7–55 (23)	L (3)	Solitary (6)	Yes (2)	Surgical (1) Surgical (7)	Size reduced (1) Disappeared (7)	26–123 (89)
				M (5)					R (4)	Multiple (1)	No (5)			
3	Nigam et al. [12]	2019	17	F (6)	2–62 (45)	FE (11), AP (7), WE (4), WL (5),	CH (4)	24–115 (56.5)	L (6)	Solitary (14)	No (17)	Antibiotics (9)	Disappeared (15)	2–40 (23)
				M (11)		VO (4), JA (4), NS (1)			R (11)	Multiple (3)		Antibiotics + Antinfiammatory (2)	Expire (2)	
												Surgical (6)		
4	Patel et al. [13]	2018	1	M	48	AP, VO		96	R	Solitary	Yes	Observation	Disappeared	12
5	Miyajima et al. [14]	2017	1	F	50	AP, VO		60	L	Solitary	Yes	Corticosteroids	Size reduced	40
6	Al-Hussaini et al. [15]	2015	1	M	8	FE, WL		80	R	Solitary	Yes	Surgical	Disappeared	4
7	Horiguchi et al. [16]	2012	1	M	76	NS		15	L	Multiple	Yes	Corticosteroids	Disappeared	4
8	Ntinas et al. [17]	2011	1	M	55	AP, WL		40	R	Solitary	No	Surgical	Disappeared	12
9	Manolaki et al. [18]	2009	1	F	9	FE, AP, WL		35	L	Solitary	Yes	Surgical	Disappeared	36
10	Tsou et al. [19]	2007	8	F (4)	28–78 (61.5)	FE (3), AP (5), WE (1), WL (3), JA	CH (2),	25–150 (73.5)	L (5)	Solitary (6)	Yes (5)	Observation (1)	Disappeared (4)	1–48 (14)
				M (4)		(2), NS (2)	GC (1)		R (2)	Multiple (2)	No (3)	Antibiotics (2)	Size reduced (3)	
									Both (1)			Paracetamol (1)	Expire (1)	
												Antibiotics—failed → Surgical (1)		
												Surgical (3)		
11	Yamaguchi et al. [20]	2007	2	F (1)	51–57 (54)	FE (1), AP (1), WL (1), NS (1)		44–51 (47.5)	L (1)	Solitary (1)	No (2)	Observation (2)	Disappeared (2)	2–12 (7)
				M (1)					R (1)	Multiple (1)				
12	Koea et al. [21]	2003	2	M (2)	23–58 (40.5)	FE (1), AP (1), WL (1), NS (1)		10–60 (35)	R (2)	Solitary (2)	Yes (2)	Observation (2)	Disappeared (1)	3–24 (13.5)
													Expire (1)	
13	Sakai et al. [22]	2001	1	F	2	FE		55	R	Solitary	Yes	Surgical	Disappeared	18
14	Hsaio et al. [23]	1999	1	F	2	FE, AP		55	L	Solitary	Yes	Surgical	Disappeared	8
15	Passalides et al. [24]	1996	1	F	14	FE, AP, WE		90	R	Solitary	No	Surgical	Disappeared	24
16	Loke et al. [22]	1994	1	F	2	FE, AP		55	L	Solitary	No	Surgical	Disappeared	3
17	Broughan et al. [22]	1993	1	M	13	FE, WE		100	R	Solitary	No	Surgical	Disappeared	36
18	Shek et al. [25]	1993	2	F (2)	31–35 (33)	FE (1), AP (2), WL (2)		150–200 (175)	R (2)	Solitary (2)	No (2)	Surgical (2)	Disappeared (2)	6–8 (7)
19	Hata et al. [22]	1992	2	F (1)	6–7 (6.5)	FE (2), AP (2), VO (2)		40–90 (65)	L (1)	Solitary (2)	No (2)	Surgical (2)	Disappeared (2)	72–192 (132)
				M (1)					R (1)					
20	Newbould et al. [25]	1992	1	M	3	NS		20	L	Solitary	No	Surgical	Disappeared	14
21	Andreola et al. [25]	1990	1	F	22	AP		130	R (1)	Solitary	Yes	Surgical (1)	Disappeared	40
22	Horiuchi et al. [26]	1990	3	M (3)	37–63 (57)	FE (2), AP (3), JA (1)	CH (1)	40–60 (50)	L (1)	Solitary (3)	No (3)	Observation (1)	Expire (3)	2–18 (6)
									R (2)			Antibiotics (1)		
												Surgical (1)		
23	Standiford et al. [25]	1989	1	M	77	FE, WE, WL		70	L	Solitary	Yes	Surgical	Disappeared	18
24	Levitt et al. [25]	1988	1	F	31	WE, WL		50	R	Solitary	Yes	Surgical	Disappeared	1
25	Kessler et al. [25]	1988	1	M	17	FE, AP		80	R	Solitary	No	Antibiotics—failed → Surgical	Disappeared	6
26	Collina et al. [27]	1987	2	M (2)	53–72 (62.5)	FE (2), WE (1)		35–70 (52.5)	R (2)	Solitary (2)	No (2)	Surgical (2)	Disappeared (1)	1–12 (6.5)
													Expire (1)	
27	Anthony and Telesinghe	1986	5	F (1)	10–61 (44)	FE (2), AP (3), VO (2), JA (2)	CH (2)	20–90 (30)	R (2)	Solitary (2)	Yes (3)	Corticosteroids (1)	Disappeared (5)	2–96 (48)
	[28]			M (4)					Both (3)	Multiple (3)	No (2)	Antibiotics—failed → Surgical (1)		
												Surgical (3)		
28	Chen [28]	1984	1	M	29	FE, WL, VO		60	R	Solitary	Yes	Surgical	Disappeared	8
29	Someren [28]	1978	1	M	4	FE, AP, WE, WL, VO		80	R	Solitary	Yes	Surgical	Disappeared	6
30	Pack and Baker [1]	1953	1	M	40	FE, WE, WL		25	R	Solitary	Yes	Surgical	Disappeared	1

*FE* fever, *AP* abdominal pain, *WE* weakness, *WL* weight loss, *VO* vomiting, *JA* jaundice, *NS* no symptoms, *CH* Cholelithiasis, *GC* gallbladder cancer.

**Table 2 cancers-16-02926-t002:** Summary of symptoms of patients affected by IPTL.

Symptoms	n (%)
Abdominal pain	38 (53.5%)
Fever	37 (52.1%)
Weight loss	20 (28.1%)
Weakness	12 (16.9%)
Vomiting	12 (16.9%)
Jaundice	9 (12.7%)

**Table 3 cancers-16-02926-t003:** Summary of treatment options in two different eras, before and after 2000.

Treatment	≤2000 n (%)	>2000 n (%)
Surgical	23 (88.5%)	22 (48.9%)
Conservative	3 (11.5%)	23 (51.1%)
Observation	1 (3.8%)	6 (13.3%)
Medical	2 (7.7%)	17 (37.7%)
Antibiotics	1 (3.8%)	11 (24.4%)
Antibioticand anti-inflammatory	-	2 (4.4%)
Paracetamol	-	1 (2.2%)
Corticosteroids	1 (3.8%)	3 (6.7%)
Total	26	45

## Data Availability

The datasets used or analyzed during the current study are available from the corresponding author upon reasonable request.

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
