# Peer review of "Inflammatory Pseudotumor of the Liver or Intrahepatic Cholangiocarcinoma, That’s the Question: A Review of the Literature"

_cancers, 2024, doi:10.3390/cancers16172926_

Round 1

Reviewer 1 Report

Comments and Suggestions for Authors

Hepatic Inflammatory Pseudotumor (IPT) is a rare disease often masquerading as a malignant tumor, resulting in misdiagnosis and unnecessary surgical resection.  Active histological diagnosis is warranted for patients with suspected IPT to avoid surgical treatment. However, IPT treatment is not yet standardized. Barabino et al. report two cases of IPT and briefly review the limited literature available on the field. This reviewer considers this manuscript should be published as a Case Report in a more specific journal on hepatology.  On the other hand, a recent review has been recently published on this topic (see PMID: 37685395). In any case, there are several issues in this manuscript at present stage that should be addressed by the authors:

- The manuscript needs extensive revision for language and grammar.

- Include a Figure with pre-operative hematoxylin and eosin staining and polyclonal for Kappa and Lambda of tissue biopsy of Case 1.

- Figure 6; specify the type of staining in sample C. Include scale bars in B and C.

- Consider including the following case described in PMID: 38745658.

Minor:

- Line 21; define US abbreviation.

- Line 67; include the abbreviation MDT.

Typo and text errors:

- Line 16; fentity => entity

- Line 167; die = > dye

- Line 244; Us => US

- Line 279 Ca => CA

Comments on the Quality of English Language

Extensive editing of English language required, e.g., rewrite sentences in Lines 17; “…to the worst one intrahepatic…”, 28; “...the make diagnosis…”, 49; “…of whom liver in 8%...”, 100; “…abdominal ultrasound mentioned...”, 152; “…resection with en bloc...”, 183; “…even if some author reported...”, 191; “…this item...”, 202; “…cases. not specific…”, 204; “…appearing as measurement not routinely…”, 243 “At regard...”

Author Response

Comments and Suggestions for Authors

  1. Hepatic Inflammatory Pseudotumor (IPT) is a rare disease often masquerading as a malignant tumor, resulting in misdiagnosis and unnecessary surgical resection.  Active histological diagnosis is warranted for patients with suspected IPT to avoid surgical treatment. However, IPT treatment is not yet standardized. Barabino et al. report two cases of IPT and briefly review the limited literature available on the field. This reviewer considers this manuscript should be published as a Case Report in a more specific journal on hepatology. On the other hand, a recent review has been recently published on this topic (see PMID: 37685395).

In any case, there are several issues in this manuscript at present stage that should be addressed by the authors:

  1. The manuscript needs extensive revision for language and grammar.
  2. Include a Figure with pre-operative hematoxylin and eosin staining and polyclonal for Kappa and Lambda of tissue biopsy of Case 1.
  3. Figure 6; specify the type of staining in sample C. Include scale bars in B and C.
  4. Consider including the following case described in PMID: 38745658.
  5. Minor:

- Line 21; define US abbreviation.

- Line 67; include the abbreviation MDT. Typo and text errors: - Line 16; fentity => entity. - Line 167; die = > dye. - Line 244; Us => US  Line 279 Ca => CA

Comments on the Quality of English Language

  1. Extensive editing of English language required, e.g., rewrite sentences in Lines 17; “…to the worst one intrahepatic…”, 28; “...the make diagnosis…”, 49; “…of whom liver in 8%...”, 100; “…abdominal ultrasound mentioned...”, 152; “…resection with en bloc...”, 183; “…even if some author reported...”, 191; “…this item...”, 202; “…cases. not specific…”, 204; “…appearing as measurement not routinely…”, 243 “At regard...”

Answer

  1. Thank you so much for the opportunity to discuss our manuscript with you, our article reported two cases of a rare disease and a systemic review of the literature. We included in the references the review suggested.

  1. We improved the English language of the paper.

  1. We are sorry but we do not have images of the biopsy performed in the Case 1. However we have reported in the article the images of the final histological examination.

  1. We improved the legend of Figure 6.

  1. The case suggested is only a case report of 3-year old boy, it is irrelevant for the issue of our review: the management of this rare disease in adult patient.

  1. We modified these errors.

  1. including the rephrasing of numerous paragraphs and some typos.

Answer about the Quality of English Language:

We have revised thoroughly the language with the assistant of an experienced scientific writer.

Reviewer 2 Report

Comments and Suggestions for Authors

This paper is very interesting, about a particular setting of liver imaging.

It is well written with a sufficient bibliography.

The issue is the "inflammatory pseudotumor" of the liver, a lesion not so rarely found in abdominal ultrasound if we know that.

The work is about diagnosis and treatment.

It starts from 2 case reports (with good images of radiological finding and histology) and it follows a review of the literature about follow-up and treatments.

Very useful for the clinicians.

Please review the quality and definitions of table 2 and 3

Reviewer 2 Report

Questions:

  1. This paper is very interesting, about a particular setting of liver imaging. It is well written with a sufficient bibliography. The issue is the "inflammatory pseudotumor" of the liver, a lesion not so rarely found in abdominal ultrasound if we know that. The work is about diagnosis and treatment. It starts from 2 case reports (with good images of radiological finding and histology) and it follows a review of the literature about follow-up and treatments. Very useful for the clinicians. Please review the quality and definitions of table 2 and 3.

Answer:

Thanks for your compliments. We improved the quality and definitions of table 2 and 3.

Reviewer 3 Report

Comments and Suggestions for Authors

The presented article is a combination of the case report and literature review.

The present form needs to be improved before publication. As for instances:

(1) The introduction is very limited and lacks information. A detailed background related to Inflammatory pseudotumor of the liver has to be incorporated.

(2) What are the reported imaging agents available to trace IPTL? Is there a selective radiotracer reported to differentiate IPTL from malignancy?

(3) What cell surface biomarkers differentiate IPTL from malignancy or ICC? What are the microenvironment parameters or markers available in IPTL?

(4) What is the status of FAP concentration in IPTL? Is it possible to target FAP in IPTL?

(5) Is any PET radiotracer available for IPTL diagnosis other than Biopsy? 

(6) Is there any report for IPTL metastasis?  

Comments on the Quality of English Language

Minor editing of the English language is required to increase fluency. For example, in Line 272, "that has to be differentiated from the bad one ICC" needs to be refreshed.

Reviewer 3 Report

Comments and Suggestions for Authors

The presented article is a combination of the case report and literature review. The present form needs to be improved before publication. As for instances:

(1) The introduction is very limited and lacks information. A detailed background related to Inflammatory pseudotumor of the liver has to be incorporated.

(2) What are the reported imaging agents available to trace IPTL? Is there a selective radiotracer reported to differentiate IPTL from malignancy?

(3) What cell surface biomarkers differentiate IPTL from malignancy or ICC? What are the microenvironment parameters or markers available in IPTL?

(4) What is the status of FAP concentration in IPTL? Is it possible to target FAP in IPTL?

(5) Is any PET radiotracer available for IPTL diagnosis other than Biopsy?

(6) Is there any report for IPTL metastasis? 

Comments on the Quality of English Language

Minor editing of the English language is required to increase fluency. For example, in Line 272, "that has to be differentiated from the bad one ICC" needs to be refreshed.

Answer:

Thank you for this comment.

  1. We improved the introduction.
  2. It is very difficult to identify the IPTL preoperatively, we suggest to perform liver biopsy for obtain a diagnosis.
  3. Currently there is no evidence on the usefulness of specific biomarkers to differentiate IPTL from tumor such as ICC or HCC.
  4. Liver tumor markers (AFP and CA 19.9) were almost within normal limits in the reported cases. There were only rare cases of IPT with elevated serum levels of AFP. In these cases, hepatic IPT mimicking an HCC.
  5. Positron emission tomography (PET) is not commonly performed; however, only few IPT cases showed abnormal metabolic activity on FDG-PET mimicking an HCC.
  6. No Hepatic inflammatory pseudotumor (IPTL) is a benign, non-metastasizing tumour.

Answer about the Quality of English Language:

We have revised thoroughly the English Language.

Round 2

Reviewer 1 Report

Comments and Suggestions for Authors

The authors have reasonably addressed the issues raised in my previous review.

Comments on the Quality of English Language

Minor editing of English language required.